# Antiparallel protocadherin homodimers use distinct affinity- and specificity-mediating regions in cadherin repeats 1-4

John M Nicoludis[1,2], Bennett E Vogt[2], Anna G Green[3], Charlotta PI Schärfe[3,4], Debora S Marks[3], Rachelle Gaudet[2]*

[1]Department of Chemistry and Chemical Biology, Harvard University, Cambridge, United States; [2]Department of Molecular and Cellular Biology, Harvard University, Cambridge, United States; [3]Department of Systems Biology, Harvard Medical School, Boston, United States; [4]Applied Bioinformatics, Department of Computer Science, University of Tübingen, Tübingen, Germany

**Abstract** Protocadherins (Pcdhs) are cell adhesion and signaling proteins used by neurons to develop and maintain neuronal networks, relying on *trans* homophilic interactions between their extracellular cadherin (EC) repeat domains. We present the structure of the antiparallel EC1-4 homodimer of human PcdhγB3, a member of the γ subfamily of clustered Pcdhs. Structure and sequence comparisons of α, β, and γ clustered Pcdh isoforms illustrate that subfamilies encode specificity in distinct ways through diversification of loop region structure and composition in EC2 and EC3, which contains isoform-specific conservation of primarily polar residues. In contrast, the EC1/EC4 interface comprises hydrophobic interactions that provide non-selective dimerization affinity. Using sequence coevolution analysis, we found evidence for a similar antiparallel EC1-4 interaction in non-clustered Pcdh families. We thus deduce that the EC1-4 antiparallel homodimer is a general interaction strategy that evolved before the divergence of these distinct protocadherin families.

*For correspondence: gaudet@mcb.harvard.edu

**Competing interests:** The authors declare that no competing interests exist.

## Introduction

Protocadherins (Pcdhs) encompass about 70% of the cadherin superfamily in mammals, and are involved in cell adhesion in the nervous system of higher animals (*Hulpiau and van Roy, 2011*; *Hulpiau et al., 2013*; *Keeler et al., 2015*; *Sotomayor et al., 2014*). Pcdhs segregate into two groups: clustered (expressed from a large gene cluster) and non-clustered. Clustered Pcdhs, comprising the α, β, and γ subfamilies, mediate neuronal survival, self-avoidance and self/nonself discrimination, and promote dendritic arborization in neuronal cells (*Emond and Jontes, 2008*; *Garrett et al., 2012*; *Kostadinov and Sanes, 2015*; *Ledderose et al., 2013*; *Lefebvre et al., 2012*; *Molumby et al., 2016*; *Suo et al., 2012*; *Wang et al., 2002*; *Weiner et al., 2005*). Clustered Pcdhs thus function analogously to insect Dscam isoforms to mediate neuronal identity (*Zipursky and Sanes, 2010*). Clustered Pcdhs are also broadly involved in synapse formation and maintenance, neuronal connectivity and neuronal survival (*Hayashi and Takeichi, 2015*; *Keeler et al., 2015*). Non-clustered Pcdhs play key roles in neuronal development (*Keeler et al., 2015*; *Kim et al., 2011*). For example, Pcdh7 is involved in germ layer differentiation (*Rashid et al., 2006*; *Yoshida, 2003*), and Pcdh1 and Pcdh8 mediate cell sorting and migration during gastrulation (*Kim et al., 1998*; *Kuroda et al., 2002*). Both clustered and non-clustered Pcdhs control these phenotypes through homophilic interactions of their extracellular cadherin (EC) repeat domains (*Hirano et al., 1999*;

*Hoshina et al., 2013*; *Kim et al., 1998*; *Kuroda et al., 2002*; *Schreiner and Weiner, 2010*; *Thu et al., 2014*; *Yamagata et al., 1999*; *Yoshida, 2003*).

Clustered Pcdh subfamilies show distinct phenotypes. In zebrafish, α and γ Pcdhs express in overlapping but distinct brain regions (*Biswas et al., 2012*). In mammals, α Pcdhs regulate sorting of olfactory sensory neuron axons into glomeruli, serotonergic axon maturation, and dendritic patterning in CA1 pyramidal neurons (*Hasegawa et al., 2012*, *2008*; *Katori et al., 2009*; *Suo et al., 2012*). The γ subfamily is important for self/non-self discrimination in retinal starburst amacrine cells and Purkinje neurons (*Kostadinov and Sanes, 2015*; *Lefebvre et al., 2012*). Thus, available data suggest that the different Pcdh subfamilies may function independently or cooperatively, perhaps depending on the brain region and/or neuronal cell type.

Our recent PcdhγA1 and PcdhγC3 EC1-3 structures revealed dimer interactions between EC2 and EC3 (*Nicoludis et al., 2015*), consistent with previous biochemical and bioinformatics data (*Schreiner and Weiner, 2010*; *Wu, 2005*). Using sequence co-evolution analysis, we predicted inter-subunit EC1-EC4 interactions, and proposed that clustered Pcdhs form extended antiparallel homodimers engaging EC1-4. A complementary biochemical and structural study arrived at a very similar docking model (*Rubinstein et al., 2015*), which was recently confirmed for α and β clustered Pcdhs (*Goodman et al., 2016*).

We determined the crystal structure of PcdhγB3 EC1-4, the first full antiparallel dimer for a γ isoform. We analyzed the clustered Pcdhs structures in light of biological, biochemical and evolutionary data to further resolve how clustered Pcdhs encode specificity. We describe how structural differences between the α, β and γ subfamilies generate distinct modes of specificity encoding. We also provide evidence that the EC1/EC4 and EC2/EC3 interfaces are functionally different: EC1/EC4

provides nonselective dimerization affinity while EC2/EC3 is generally responsible for enforcing specificity. Finally, we extend our sequence coevolution analysis to the non-clustered Pcdhs and provide evidence that the EC1-4 interaction is broadly used by Pcdhs.

## Results and discussion

### Structure of the PcdhγB3 EC1-4 extended antiparallel dimer

The in vitro-refolded recombinantly-expressed PcdhγB3 EC1-4 (47 kDa) yielded two peaks on size exclusion chromatography (SEC; *Figure 1—figure supplement 1*). Based on multi-angle light scattering (MALS; *Figure 1—figure supplement 1*), peak 1 was wide and polydisperse (~200–300 kDa). Peak 2 was monodisperse at 80 kDa – consistent with a dimer – and readily yielded a crystal structure (*Figure 1—source data 1*). As expected, each EC forms a seven-stranded Greek key β-sandwich motif (*Figure 1A*), similarly to other clustered Pcdh structures (*Goodman et al., 2016*; *Nicoludis et al., 2015*; *Rubinstein et al., 2015*). Notably, EC4 has a unique β-strand arrangement compared to EC1-EC3 (*Figure 1B,C*) and all known cadherin repeat structures. Strand β1a is extended by 4–5 residues, while β1b is correspondingly shortened. Additionally, while in EC1-3 strand β2 splits into β2a and β2b, interacting with strands β5 and β1a, respectively, in EC4 it forms a continuous strand interacting with both simultaneously. This distinct structural feature contributes to intersubunit EC1/EC4 interactions (see below).

Although the asymmetric unit contains a single PcdhγB3 molecule, a crystallographic two-fold axis generates an antiparallel dimer with intersubunit EC1/EC4 and EC2/EC3 interactions (*Figure 1A*). This dimer is consistent with the PcdhγA1 EC1-3 crystal structure, validating the previously predicted interface (*Nicoludis et al., 2015*; *Rubinstein et al., 2015*), and with recent α and β Pcdhs structures (*Goodman et al., 2016*), confirming that this interaction mechanism is conserved among all clustered Pcdh subfamilies (*Figure 1D*). The structures do differ noticeably in overall twist, including subfamily-specific differences in relative EC1/EC4 orientation (*Figure 1E*).

The linear architecture of clustered Pcdhs enables extended antiparallel dimer interfaces. Overall, the tilt and azimuthal angles between adjacent clustered Pcdh repeats are distinct from those of classical cadherins (*Figure 1—figure supplement 2*) (*Nicoludis et al., 2015*). Classical cadherins, which typically dimerize through EC1/EC1 interfaces, exhibit smaller tilt angles and thus an overall curved structure (*Boggon et al., 2002*). Notably, the clustered Pcdh repeat orientation is such that EC1 and EC3 use the same face for intersubunit contacts, as do EC2 and EC4 (*Figure 1—figure supplement 3*), suggesting that longer cadherins could readily form even more extended interfaces.

### Clustered protocadherin subfamilies have distinct specificity mechanisms dictated by structural differences

Clustered Pcdh subfamilies control different phenotypes in vivo and have discrete expression patterns (*Biswas et al., 2012*; *Keeler et al., 2015*), suggesting that they encode specificity using distinct modes, which may relate to subfamily-specific structural features. To investigate this hypothesis, we calculated the isoform conservation ratio (ICR) within individual subfamilies, which quantifies the extent to which individual residue positions are conserved among orthologs (same isoform in different species) and diversified in paralogs (different isoforms in the same species) (*Nicoludis et al., 2015*), resulting in three ICR value sets for the α, β and γ subfamilies, respectively (*Figure 2—figure supplement 1*). To account for subfamily differences in sequence conservation, we normalized the ICR values by dividing by the subfamily average. We then mapped them onto the Pcdhα7 EC1-5, Pcdhβ8 EC1-4 and PcdhγB3 EC1-4 structures (*Figure 2A*). Comparing the structures and isoform-specific conservation in the different subfamilies allowed us to identify key specificity determinant regions for individual subfamilies. We illustrate three examples of how the subfamilies have encoded specificity using unique structural features.

In α isoforms, the EC2 β4-β5 loop is enriched in high-ICR and chemically diverse residues, and differs in conformation in the Pcdhα4 and Pcdhα7 structures (*Figure 2B*): the Pcdhα4 EC2 β4-β5 loop contacts β1b of EC3, while the corresponding loop in Pcdhα7 does not, suggesting variable interactions in other isoforms. In comparison, the EC2 β4-β5 loop residues in both β and γ isoforms have lower ICR values, more similar loop structure, and do not contact β1b of EC3. Thus, this loop may have evolved to generate diversity within α isoforms, but not in other subfamilies.

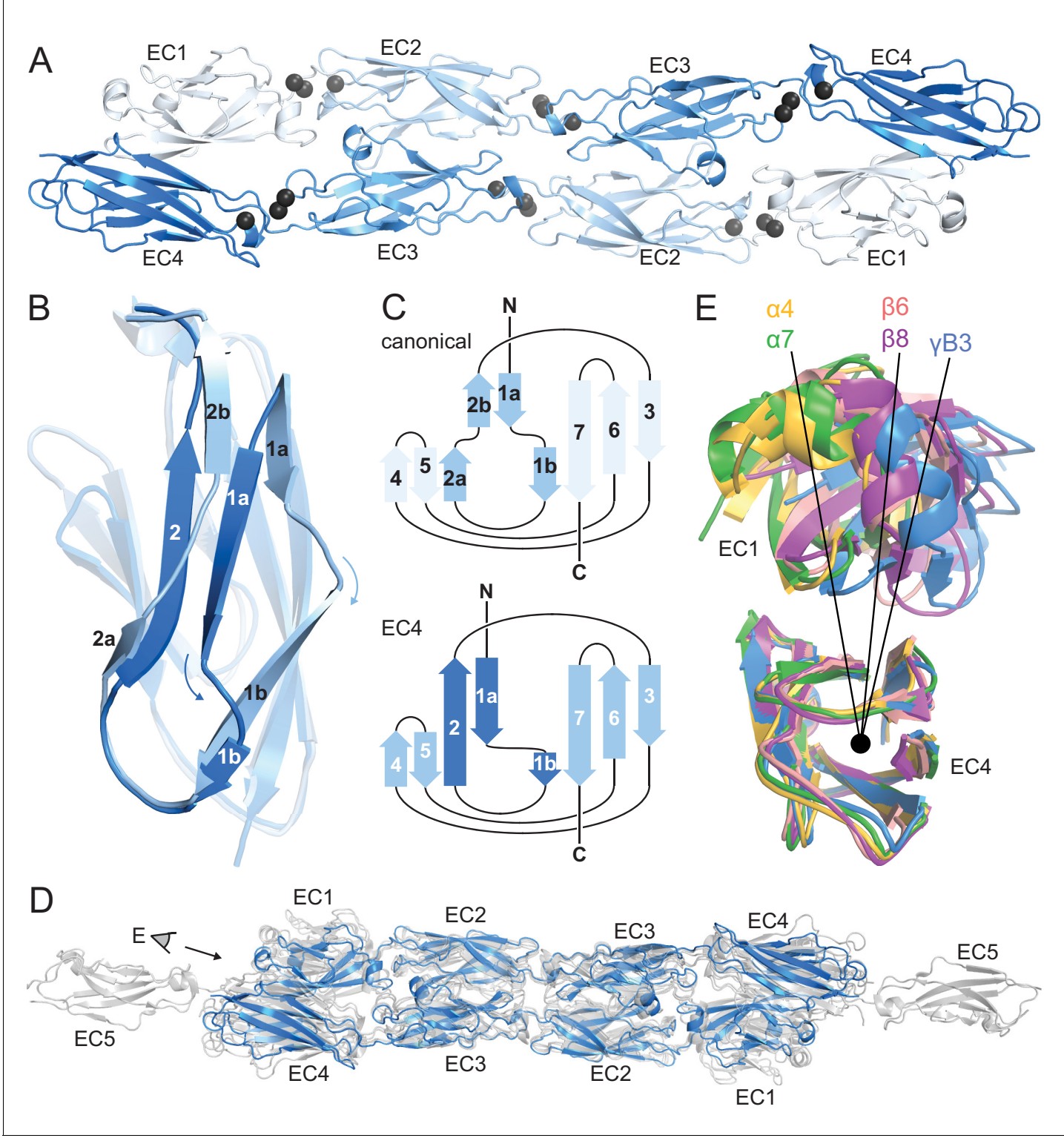

**Figure 1.** PcdhγB3 EC1-4 extended antiparallel dimer relies on unusual EC4 β-strand arrangement and is similar to other clustered Pcdh dimers. (**A**) Structure of the PcdhγB3 EC1-4 antiparallel dimer, with each EC a different shade of blue and the $Ca^{2+}$ ions in grey. (**B**) Superposition of PcdhγB3 EC2 and EC4 highlighting the differences in β-strands 1 and 2. (**C**) Comparison of the canonical cadherin (top) and EC4 (bottom) β-strand arrangement. (**D**) The structures of Pcdh dimers α4 EC1-4, α7 EC1-5, β6 EC1-4, and β8 EC1-4 (grey) were superimposed using the dimeric EC2-3 region onto γB3 EC1-4 (blue), illustrating variations in twist/corkscrew. (**E**) The EC4 domains of clustered Pcdh structures (colored as labeled) were superimposed, highlighting EC1 position differences that correlate with subfamily. Point of view (eye symbol) shown in (**D**). See *Figure 1—figure supplements 1–4*.

*Figure 1 continued on next page*

*Figure 1 continued*

The following source data and figure supplements are available for figure 1:

**Source data 1.** Statistics for PcdhγB3 EC1-4 structure.

**Figure supplement 1.** PcdhγB3 refolding yields two species, one of which corresponds to monodisperse dimeric protein.

**Figure supplement 2.** Protocadherins and non-classical cadherins have a distribution of orientation between repeat pairs that is distinct from classical cadherins.

**Figure supplement 3.** EC1 and EC3 use the same face for intersubunit contacts, as do EC2 and EC4.

**Figure supplement 4.** HEPES molecule near the EC2/EC3 interface.

In β isoforms, the Phe-$X_{10}$-Phe loop between β3 and β4 of EC3 has limited diversity compared to α and γ isoforms and wedges between the EC2 β4-β5 loop and β2b strand (*Figure 2C*). In contrast, the Phe-$X_{10}$-Phe loop of α and γ isoforms has a helical conformation, and has residues with higher ICR values and greater chemical diversity. Therefore alterations in secondary structure can affect how specificity is encoded within the subfamilies.

The short loop following the extended β1a strand in EC4 contacts the EC1 β6-β7 loop (*Figure 2D*), and there are large structural differences in the EC1/EC4 interaction between subfamilies (*Figure 1E*). α Isoforms have low-ICR residues at this interface, whereas β and γ isoforms have higher ICR value residues. This thus suggests that the large structural differences drive inter-subfamily specificity, on which may be layered additional isoform specificity.

In all cases, sequence regions with high isoform-specific conservation correlate with interface contacts, revealing the interplay between dimer structure and how subfamilies encode specificity. Diversity in the composition and conformation of loop regions provides distinct specificity mechanisms to subfamilies. Phylogenetic analysis indicates that isoforms are more similar within than across subfamilies (*Sotomayor et al., 2014*; *Wu, 2005*), and the available structures show that the interface architecture is more similar within subfamilies as well (*Figure 1E*) (*Goodman et al., 2016*; *Nicoludis et al., 2015*). With this insight, the dimer interface seen in the PcdhγC3 EC1-3 crystal structure may represent a unique dimer architecture for C-type isoforms (*Nicoludis et al., 2015*), as these isoforms are transcriptionally, functionally and evolutionarily distinct from the subfamilies in which they reside (*Chen et al., 2012*; *Frank et al., 2005*; *Kaneko et al., 2006*). Distinct expression of the clustered Pcdh subfamilies in different tissues and at different developmental stages supports the necessity for intra-subfamily specificity (*Biswas et al., 2012*; *Frank et al., 2005*). Differences in subfamily structure and isoform-specific conservation suggest that homophilic specificity mechanisms emerged independently in each subfamily through diversification of subfamily-specific interface contacts.

## The EC1/EC4 interaction provides affinity of dimerization

The EC2/EC3 interaction is integral to clustered Pcdh dimerization specificity, as evidenced by bioinformatics and cell-aggregation assays (*Nicoludis et al., 2015*; *Rubinstein et al., 2015*; *Schreiner and Weiner, 2010*; *Thu et al., 2014*; *Wu, 2005*). We sought to understand the functional purpose of the EC1/EC4 interaction, and made three observations. First, for all isoforms with available structures, fewer EC1/EC4 interface residues have high isoform-specific conservation compared to the EC2/EC3 interface residues (*Figure 2A*, *Figure 2—figure supplement 1*). Second, interface residues shared by most isoforms are more hydrophobic in EC1/EC4 than in EC2/EC3 (*Figure 3A*). Third, the PcdhγB3 EC1/EC4 interface is much larger (BSA = 976 Å$^2$ per protomer) than the EC2/EC3 interface (555 Å$^2$ per protomer). The lack of isoform specificity, the hydrophobic nature, and large interface area together suggest that the EC1/EC4 interface promotes binding with little specificity.

To probe this hypothesis, we used the Computational Interface Alanine Scanning Server to assess each interface residue's contribution to the free energy of complex formation ($\Delta\Delta G_{calc}$) when

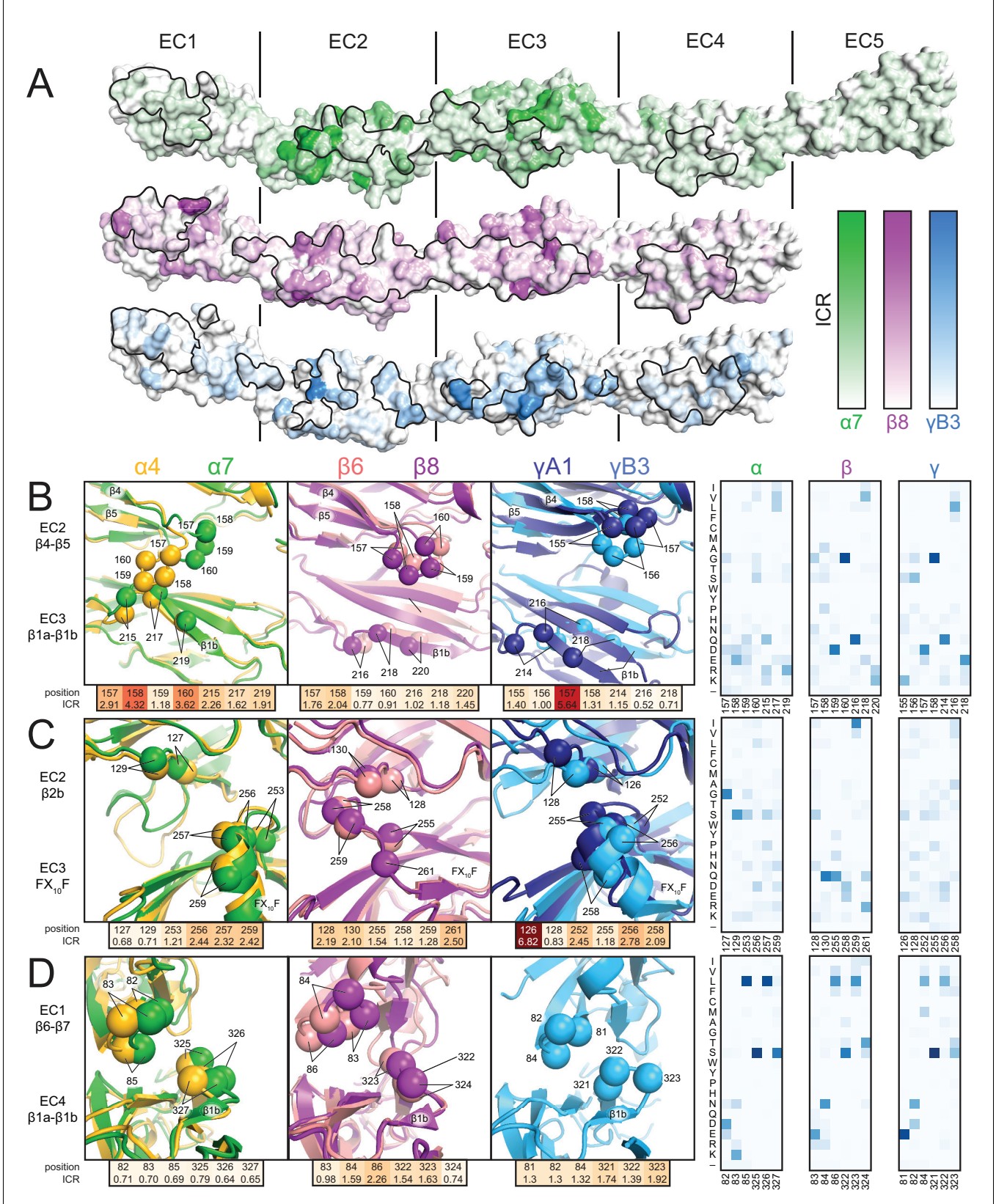

**Figure 2.** Isoform-specific conservation and structural differences reveal subfamily differences in diversity generation. (A) Subfamily-specific ICR values mapped onto the surfaces of Pcdhα7 (top, green), Pcdhβ8 (middle, magenta) and PcdhγB3 (bottom, blue). The black outline marks the dimer interface

*Figure 2 continued on next page*

*Figure 2 continued*

footprint. (B, C, D) Unique structural features of the α (left), β (center), and γ (right) structures (colored according to *Figure 1*). ICR values for highlighted residues shown below and normalized amino acid frequencies for these positions shown on the right. See *Figure 2—figure supplement 1*.

The following figure supplement is available for figure 2:

**Figure supplement 1.** Clustered Pcdh subfamilies have distinct patterns of isoform-specific conservation.

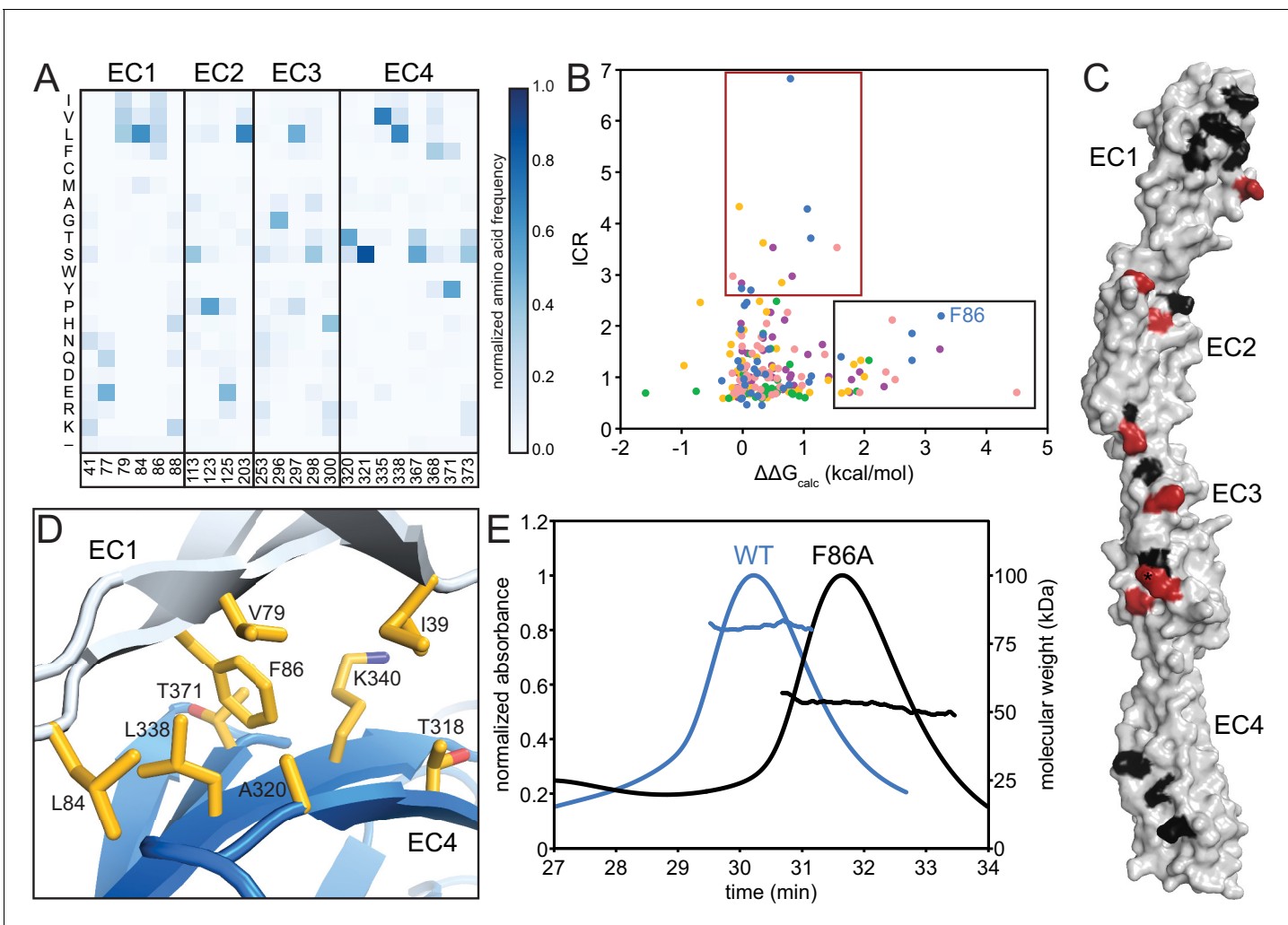

**Figure 3.** The EC1/EC4 interface is enriched in affinity-driving hydrophobic residues, while the EC2/EC3 interface contains high-ICR residues driving specificity. (A) Amino acid frequencies in clustered Pcdhs of conserved interface residues (see *Figure 1—figure supplement 2*). (B) Plot of ICR value and $\Delta\Delta G_{calc}$ of interface residues of Pcdh α4 EC1-4 (yellow), α7 EC1-5 (green), β6 EC1-4 (salmon), β8 EC1-4 (magenta), γB3 EC1-4 (blue). Two subsets of interface residues segregate from the main cluster: high-$\Delta\Delta G_{calc}$ and low-ICR residues ('affinity'; black box) and low-$\Delta\Delta G_{calc}$ and high-ICR residues ('specificity'; crimson box). Residue F86 from PcdhγB3 EC1-4 is labeled. (C) High-$\Delta\Delta G_{calc}$ and low-ICR residues (black) map primarily to EC1 and EC4, while low-$\Delta\Delta G_{calc}$ and high-ICR residues (crimson) primarily map to EC2 and EC3. N253 (*) is found in the 'specificity' region for γB3 and in the 'affinity' region for β6 and β8. (D) The EC1/EC4 interface features a hydrophobic cluster, with EC1 F86 near its center. (E) SEC-MALS profiles of WT PcdhγB3 EC1-4 (blue; molecular weight 82 kDa) and F86A (black; molecular weight 52 kDa) run on a Superdex S200 10/300 column, are consistent with dimeric and monomeric proteins, respectively.

computationally mutated to alanine (*Kortemme and Baker, 2002*; *Kortemme et al., 2004*). Using the five available EC1-4 interfaces, two residue groups emerged from comparing the ICR values to $\Delta\Delta G_{calc}$: one group with low ICR values and high $\Delta\Delta G_{calc}$, the other with high ICR values and low $\Delta\Delta G_{calc}$ (*Figure 3B*). These residue groups can be regarded as contributing to the affinity and specificity of the complex, respectively. When mapped on PcdhγB3, predicted affinity residues concentrated on EC1 and EC4, and predicted specificity residues on EC2 and EC3, corroborating the distinction between EC1/EC4 and EC2/EC3 interactions (*Figure 3C*).

In the PcdhγB3 EC1/EC4 interface, F86, one of the predicted affinity-driving residues from EC1, wedges into a cavity created by hydrophobic EC4 residues (*Figure 3D*). A PcdhγB3 EC1-4 F86A mutant indeed disrupted dimerization, resulting in a monomeric protein as measured by MALS (*Figure 3E*). Thus, the hydrophobic interactions between EC1 and EC4 are crucial to dimerization. Analogously, purified EC1-3 constructs failed to dimerize in vitro whereas EC1-4 constructs did (*Nicoludis et al., 2015*; *Rubinstein et al., 2015*), and K562 cells expressing ΔEC1 or ΔEC4-6 constructs did not aggregate while cells expressing chimeras in which EC1 and EC4 derived from different paralogs did (*Schreiner and Weiner, 2010*; *Thu et al., 2014*). Together, these results indicate that the EC1/EC4 interaction is not strictly required for the specificity of dimerization but it drives dimerization affinity through non-specific hydrophobic interactions.

## Antiparallel EC1-4 interaction is predicted in non-clustered Pcdhs

The antiparallel EC1-4 interaction architecture can encode diverse specificities within the clustered Pcdh family. Is this architecture unique to clustered Pcdhs or is it ancestral, and thus also found in non-clustered Pcdhs? These include the δ-1 (Pcdh1, Pcdh7, Pcdh9, Pcdh11) and δ-2 (Pcdh8, Pcdh10, Pcdh17, Pcdh18, Pcdh19) families that are integral to the development and maintenance of the nervous system (*Keeler et al., 2015*; *Kim et al., 2011*). We used sequence coevolution analysis, which successfully predicted the clustered Pcdh interface (*Nicoludis et al., 2015*) (*Figure 4—figure supplement 1*),to look for evidence of an antiparallel interface in non-clustered Pcdhs (*Figure 4A*). As in clustered Pcdhs, most covarying residue pairs in non-clustered Pcdhs were intra-domain structural contacts of the well-conserved cadherin fold. Additionally, several covarying pairs are found between EC2 and EC3, or EC1 and EC4, similar to those observed for the clustered Pcdhs. When mapped onto the PcdhγB3 dimer, these covarying pairs are somewhat further apart than true interface contacts (*Figure 4B*), which could be due to differences in dimerization interfaces, as we observe between the clustered Pcdh families, or in the δ-1 or δ-2 Pcdhs secondary structure, for which there are no available structures. This analysis thus predicts an antiparallel EC1-4 interaction in members of the non-clustered Pcdhs. Notably, we cannot determine whether all members or only a subset – and if so, which – likely use this architecture. However maximum parsimony suggests that the ancestral Pcdh used the antiparallel EC1-4 dimer interaction, and Pcdh members which do not show this interaction mechanism either diverged before it evolved or lost it subsequently.

Finally, we looked at the composition of a predicted non-clustered Pcdh interface, by selecting residues homologous to those found at clustered Pcdh interfaces. The predicted EC1/EC4 interface residues are predominantly hydrophobic, while EC2/EC3 residues have more polar and ionic character (*Figure 4C*). Notably, positions 41 and 77 in EC1 and 320, 321, 371 and 373 in EC4 are more hydrophobic in non-clustered than clustered Pcdhs, indicating that these may form contacts in some non-clustered Pcdhs. Thus, like in the clustered Pcdhs, the EC1/EC4 interaction may promote dimer affinity while the EC2/EC3 interaction provides specificity.

## Conclusions

Recently, we and others predicted that clustered Pcdhs form homophilic antiparallel EC1-EC4 complexes based on crystal structures, mutagenesis and bioinformatics (*Nicoludis et al., 2015*; *Rubinstein et al., 2015*). Our structure of PcdhγB3 EC1-4 confirms our sequence coevolution analysis, demonstrating the robustness of the analysis and revealing the molecular details of this interaction. Here we extended this prediction to other non-clustered Pcdhs using sequence coevolution analysis.

Analysis of the PcdhγB3 EC1-4 structure in comparison to α or β subfamily dimers revealed structural differences that correlated with differences in isoform-specific conservation, indicating distinct specificity mechanisms. Unlike the Dscams, where isoforms vary at specific alternatively-spliced

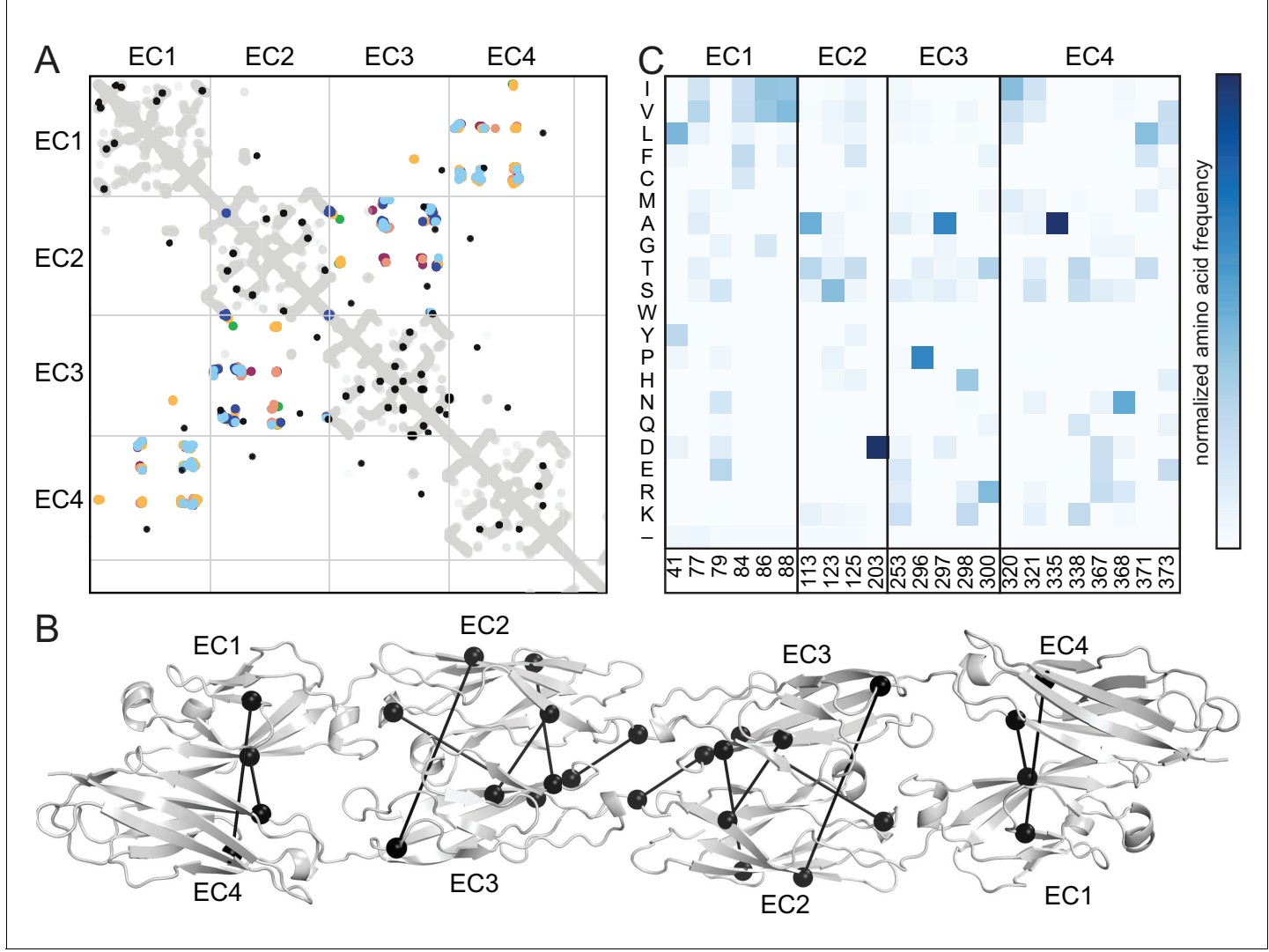

**Figure 4.** Evolutionary couplings in non-clustered Pcdhs predict an antiparallel interface engaging EC1-EC4. (A) The top 38 covarying pairs are shown in black, and include a number of EC1-EC4 and EC2-EC3 covarying residue pairs. The intramolecular contact maps of PcdhγB3 EC1-4, Pcdhα4 EC1-4, Pcdhα7 EC1-4, Pcdhβ6 EC1-4, Pcdhβ8 EC1-4 and PcdhγA1 EC1-3 are in gray for reference. The observed interface contact residues are also mapped (α4, yellow; α7, green; β6, salmon; β8, magenta; γB3, blue; γA1, dark blue). (B) Covarying residue pairs across EC1-EC4 or EC2-EC3 are mapped onto the PcdhγB3 EC1-4 structure with a line between coupled residues. Alignments and evolutionary couplings in *Figure 4—source data 1* and *2*. (C) Amino acid frequencies at non-clustered Pcdh alignment positions corresponding to the conserved interface residue positions identified in clustered Pcdhs (*Figure 1—figure supplement 2*). See *Figure 4—figure supplements 1* and *2*.

The following source data and figure supplements are available for figure 4:

**Source data 1.** Alignment of non-clustered Pcdhs EC1-4.

**Source data 2.** Evolutionary couplings from the non-clustered Pcdh alignment.

**Source data 3.** Alignment of clustered Pcdhs EC1-4.

**Source data 4.** Evolutionary couplings from the clustered Pcdh alignment.

**Figure supplement 1.** Evolutionary couplings in clustered Pcdhs are consistent with all available EC1-EC4 antiparallel homodimeric interfaces.

**Figure supplement 2.** Phylogenetic tree distinguishes clustered from non-clustered Pcdhs.

regions (*Li et al., 2016*; *Meijers et al., 2007*; *Sawaya et al., 2008*; *Wojtowicz et al., 2007*), the clustered Pcdh subfamilies are structurally diverse, and thus can encode specificity in different ways.

We identified a hydrophobic interaction between EC1 and EC4 that contributes to dimerization affinity, whereas its conservation among clustered Pcdh isoforms suggests that this interaction is not a driver of specificity. Overall, our data support a general role for conserved hydrophobic EC1/EC4 interactions in affinity, and for highly diversified polar EC2/EC3 contacts in specificity, and sequence analyses suggest that this is conserved in at least some non-clustered Pcdhs.

## Materials and methods

### Expression, purification and crystallization of PcdhγB3 EC1-4

Human PcdhγB3 EC1-4 (residues 1–414, not counting the signal peptide) was cloned into pET21 with a C-terminal hexahistidine tag, expressed in BL21 Gold (DE3) *Escherichia coli* cells in terrific broth. Cells were induced at $OD_{600}$ = 0.8 with 0.5 mM isopropyl β-D-1-thiogalactopyranoside (IPTG) at 37°C for 4 hr, harvested and lysed by sonication in 8 M guanadinium hydrochloride (GuHCl), 50 mM HEPES pH 7.5, 2 mM $CaCl_2$, and 20 mM imidazole. Cell lysates were diluted to 5 M GuHCl and loaded onto Ni-Sepharose, washed with 50 mM HEPES pH 7.5, 250 mM NaCl, 10 mM $CaCl_2$, and 25 mM imidazole and eluted with 250 mM imidazole. Eluted protein was refolded at 1 mg/mL in 12 hr dialysis steps reducing the GuHCl concentration from 2.5 M to 1.25 M and finally 0 M in refolding buffer (100 mM Tris pH 8.5, 10 mM $CaCl_2$, 1 mM EDTA, 5 mM dithiothreitol (DTT), and 0.5 M L-arginine). Concentrated refolded protein was purified by size-exclusion chromatography (SEC) on a Superdex 200 16/60 column (GE Healthcare, Pittsburgh, PA) in 20 mM Tris pH 8.5, 200 mM NaCl, and 2 mM $CaCl_2$ (SEC buffer). Two peaks were isolated and each peak was run again separately by SEC before being concentrated for crystallization.

### Multi-angle light scattering (MALS)

Approximate molecular mass of PcdhγB3 EC1-4 protein (WT or F86A mutant) was determined using a Superdex S200 10/300 column (GE Healthcare, Pittsburgh, PA) with in-line Wyatt Dawn Heleos II and Optilab T-rex refractive index detectors. Protein (100 μL at 4 mg/mL) was injected and run at 0.4 mL/min in SEC buffer. Signals were aligned, normalized and band-broadened using bovine serum albumin as a standard.

### Crystallization, data collection, and structure determination and analysis

Crystals were obtained by vapor diffusion at room temperature in 0.1 M HEPES pH 7, 4% ethylene glycol, and 5% polyethylene glycol monomethyl ether 500 in a 0.3 μL protein (12 mg/mL) to 0.3 μL reservoir drop, then cryoprotected with reservoir with 20% glycerol before flash cooling in liquid $N_2$. Diffraction data (*Figure 1—source data 1*) were processed in HKL2000 (*Otwinowski and Minor, 1997*). The PcdhγB3 EC1-4 structure was determined by an iterative molecular replacement search with single domains of the PcdhγA1 EC1-3 structure (PDBID 4zi9) in PHENIX (*Adams et al., 2010*). Model building was done in Coot (*Emsley and Cowtan, 2004*) and refinement in PHENIX (*Adams et al., 2010*). We analyzed the physicochemical properties of the dimer interface using PISA (*Krissinel and Henrick, 2007*). In the structure, we found a HEPES molecule near the EC2/EC3 interface that forms a salt bridge with N253 and N155 (*Figure 1—figure supplement 4*). MALS data were collected with Tris as the buffer (*Figure 1—figure supplement 1*), indicating that HEPES is not required for dimerization.

### ICR value calculations

Overall percent identity of the most common residue at each position was used to calculate ICR values, dividing the percent identity across subfamily orthologs by the percent identity across subfamily paralogs. ICR values were then normalized by dividing by the whole sequence ICR average within each subfamily. The alignment and identity data are provided here (*Nicoludis et al., 2015*).

## Computational Interface Alanine Scanning Server

Pcdhα4 EC1-4 (5dzw), Pcdhα7 EC1-5 (5dzv), Pcdhβ6 EC1-4 (5dzx), Pcdhβ8 EC1-4 (5dzy), and PcdhγB3 EC1-4 (5k8r) dimer structures were submitted to the Computational Interface Alanine Scanning Server using default settings (*Kortemme and Baker, 2002*; *Kortemme et al., 2004*).

## Covariation analyses

Previously, we generated an alignment of clustered Pcdhs using mouse PcdhγC3 and manually filtered by phylogeny, using FastTree 2.1 (*Price et al., 2010*), to eliminate non-clustered Pcdhs (*Figure 4—figure supplement 2*) (*Nicoludis et al., 2015*). Both this clustered Pcdh and the now separated non-clustered Pcdh alignment were filtered to remove sequences with more than 50% gaps and columns with more than 30% gaps, and trimmed to contain only EC1-EC4. The clustered and non-clustered Pcdh alignments had 2660 and 405.5 effective non-redundant sequences, respectively; sequences were considered redundant and downweighted when more than 90% identical over their full length (*Hopf et al., 2014*). Evolutionary couplings (*Hopf et al., 2014*; *Marks et al., 2011*) were computed using the updated 'PLMC' algorithm (*Weinreb et al., 2015*) available on https://github.com/debbiemarkslab/plmc, which uses a pseudo maximum likelihood approximation (*Balakrishnan et al., 2011*; *Ekeberg et al., 2013*; *Kamisetty et al., 2013*). Alignments and all EC scores are provided (*Figure 4—source data 1–4*).

We used the precision of the intra-domain evolutionary couplings to determine whether the inter-domain evolutionary couplings are likely to be true. For the clustered Pcdh alignment, 83 non-local (more than five residues apart) contacts fall above a threshold of 80% precision of the intra-domain evolutionary couplings. Intra-domain evolutionary couplings are considered true if they correspond to structural contact (minimum atom distance < 8 Å) in any structure (*Figure 4—figure supplement 1*). Based on the same 80% precision threshold, the top 38 non-local ECs are significant in the non-clustered Pcdh alignment. We exclude couplings between residues greater than 400 in this analysis due to the false signal from gaps in this region.

## Acknowledgements

We thank Kelly Arnett, director of the Center for Macromolecular Interactions, Harvard Medical School) for help in collecting SEC-MALS data. We thank the beamline staff of NE-CAT at the Advanced Photon Source (Argonne, IL, USA) for help with data collection. NE-CAT is funded by NIH (P41 GM103403 and S10 RR029205) and the Advanced Photon Source by the US Department of Energy (DE-AC02-06CH11357). Evolutionary couplings analysis was conducted on the Orchestra High Performance Compute Cluster at Harvard Medical School, which is funded by the NIH (NCRR 1S10RR028832-01). Financial support (to JMN) was provided by the National Defense Science and Engineering Graduate Fellowship. AGG was supported by the National Science Foundation Graduate Research Fellowship (DGE1144152). DSM was supported by National Institutes of Health (GM106303).

## Additional information

### Funding

| Funder | Grant reference number | Author |
| --- | --- | --- |
| National Science Foundation | DGE1144152 | Anna G Green |
| National Institutes of Health | NCRR 1S10RR028832-01 | Debora S Marks |
| National Institute of General Medical Sciences | GM106303 | Debora S Marks |
| National Defense Science and Engineering Graduate Fellowship | | John M Nicoludis |
| National Institute of General Medical Sciences | P41 GM103403 | Rachelle Gaudet |
| National Institutes of Health | S10 RR029205 | Rachelle Gaudet |

The funders had no role in study design, data collection and interpretation, or the decision to submit the work for publication.

## Author contributions

JMN, AGG, Conception and design, Acquisition of data, Analysis and interpretation of data, Drafting or revising the article; BEV, Conception and design, Acquisition of data, Drafting or revising the article; CPIS, Drafting or revising the article, Contributed unpublished essential data or reagents; DSM, RG, Conception and design, Analysis and interpretation of data, Drafting or revising the article

## Author ORCIDs

Rachelle Gaudet, http://orcid.org/0000-0002-9177-054X

# Additional files

## Major datasets

The following datasets were generated:

| Author(s) | Year | Dataset title | Dataset URL | Database, license, and accessibility information |
|---|---|---|---|---|
| Nicoludis JM, Vogt BE, Gaudet R | 2016 | Structure of human clustered protocadherin gamma B3 EC1-4 | http://www.rcsb.org/pdb/explore.do?structureId=5K8R | Publicly available at the RCSB Protein Data Bank (accession no: 5K8R) |
| Nicoludis JM, Vogt BE, Gaudet R | 2016 | Structure of human clustered protocadherin gamma B3 EC1-4 | https://data.sbgrid.org/dataset/325 | Publicly available at Structural Biology Data Grid (accession no: 325) |

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
