## [Decision Letter]

Thank you for submitting your article "Antiparallel protocadherin homodimers use distinct affinity- and specificity-mediating regions in cadherin repeats 1-4" for consideration by *eLife*. Your article has been reviewed by two peer reviewers, one of whom is a member of our Board of Reviewing Editors, and the evaluation has been overseen by Arup Chakraborty as the Senior Editor. The reviewers have opted to remain anonymous.

The reviewers have discussed the reviews with one another and the Reviewing Editor has drafted this decision to help you prepare a revised submission. The reviewers and editors are in agreement that the work is worthy of publication and only have a few minor concerns that would need to be addressed before acceptance.

Summary:

The authors report the structure of the first four extracellular cadherin repeats of PcdhgB3, the first of the γ subfamily of clustered protocadherins. The structure reveals an antiparallel dimer in which the four domains mediate dimerization. Using the previously known α- and β-subfamily dimer structures, the authors provide a detailed structural analysis of dimerization specificity throughout the clustered protocadherins: it appears that the EC2-EC3 interaction contributes to the specificity of binding within each subfamily, while the EC1-EC4 interface provides for a more general nonselective binding affinity. They also predict that the non-clustered protocadherins likely work through similar mechanisms. This provides insights into the binding mechanisms of the entire clustered protocadherin family that supports and extends previous functional studies. Overall it represents a significant advance in understanding this large family of adhesion proteins.

Minor points:

1) First paragraph, Results and Discussion and Figure 1: "Topology" refers to the connectivity of secondary structure elements that interact with one another. The topology of EC4 is not different from other EC domains – it simply has a split β strand but it hydrogen bonds interacts with the same neighboring strands. It is fine to point out this difference but is not a topological difference.

2) Materials and methods section, subsection “Covariation analyses”: "…405.5 non-redundant sequences…" – what is a 0.5 sequence?

3) [Supplementary-material SD1-data] (Crystallographic statistics):a) The number of significant figures is inappropriate, please fix (true of almost every entry, e.g. unit cell lengths, R values, B factors); b) "Ligands" – please specify, this presumably means Ca^2+^, HEPES, anything else?

---

## [Author Response]

*Summary:*

*The authors report the structure of the first four extracellular cadherin repeats of PcdhgB3, the first of the γ subfamily of clustered protocadherins. The structure reveals an antiparallel dimer in which the four domains mediate dimerization. Using the previously known α- and β-subfamily dimer structures, the authors provide a detailed structural analysis of dimerization specificity throughout the clustered protocadherins: it appears that the EC2-EC3 interaction contributes to the specificity of binding within each subfamily, while the EC1-EC4 interface provides for a more general nonselective binding affinity. They also predict that the non-clustered protocadherins likely work through similar mechanisms. This provides insights into the binding mechanisms of the entire clustered protocadherin family that supports and extends previous functional studies. Overall it represents a significant advance in understanding this large family of adhesion proteins.*

*Minor points:*

*1) First paragraph, Results and Discussion and Figure 1: "Topology" refers to the connectivity of secondary structure elements that interact with one another. The topology of EC4 is not different from other EC domains – it simply has a split β strand but it hydrogen bonds interacts with the same neighboring strands. It is fine to point out this difference but is not a topological difference.*

Thank you for pointing this out. We have now rephrased our descriptions:

Within the Results and Discussion section, we now state (new wording italicised): “Notably, EC4 has a unique *β-strand arrangement* compared to EC1-EC3 (Figure 1) […] This distinct *structural feature* contributes to intersubunit EC1/EC4 interactions (see below).” In the legend to Figure 1, two instances of the word “topology” were replaced with “*β-strand arrangement*”.

*2) Materials and methods section, subsection “Covariation analyses”: "…405.5 non-redundant sequences…" – what is a 0.5 sequence?*

The numbers listed actually correspond to the effective number of sequences in the alignment. To calculate the effective number of sequences, each sequence is weighted by dividing by one plus its number of neighbors, which we define as sequences that are more than 90% identical. For example, a sequence that has one neighbor in the alignment counts as 0.5 sequence. Because each sequence will have a different number of neighbors, this can result in non-integer totals when the weighted numbers of sequences are summed. We have now clarified this in the methods, in the following reworded sentence: “The clustered and non-clustered Pcdh alignments had 2660 and 405.5 effective non-redundant sequences, respectively; sequences were considered redundant and downweighted when more than 90% identical over their full length (Hopf et al., 2014).”

*3) [Supplementary-material SD1-data] (Crystallographic statistics):a) The number of significant figures is inappropriate, please fix (true of almost every entry, e.g. unit cell lengths, R values, B factors);*

We have edited the table to reduce the number of significant digits so as to be more in line with typical statistics provided in publications.

*b) "Ligands" – please specify, this presumably means Ca^2+^, HEPES, anything else?*

We have added the list of ligands in the first column, and the corresponding number of non-hydrogen atoms in the second column, as follows:

Ligands (Ca^2+^, Cl^-^, Na^+^, HEPES, ethylene glycol) 31 (9, 2, 1, 15, 4)